# Co-Design Model of Support for Child and Family Health Nurse Practice with Culturally and Linguistically Diverse Families

**DOI:** 10.3390/ijerph21101274

**Published:** 2024-09-25

**Authors:** Mehrnoush Bonakdar Tehrani, Stacy Blythe, Suza Trajkovski, Lynn Kemp

**Affiliations:** 1Translational Research and Social Innovation (TReSI), Ingham Institute for Applied Medical Research, School of Nursing and Midwifery, Western Sydney University, Liverpool, NSW 2170, Australia; mehrnoush.bonakdar@westernsydney.edu.au; 2School of Nursing and Midwifery, Faculty of Health, University of Technology Sydney, Broadway, NSW 2007, Australia; stacy.blythe@uts.edu.au; 3School of Nursing and Midwifery, Western Sydney University, Penrith, NSW 2751, Australia; s.trajkovski@westernsydney.edu.au

**Keywords:** health service inequity, cultural diversity, child and family health nursing, programmes and services

## Abstract

Culturally and linguistically diverse (CALD) mothers with young children face multiple inequities in accessing primary health services, such as language barriers, social isolation, low health literacy, and the availability of appropriate interpretation services. These inequities are persistent and indicate that child and family health nurse (CFHN) services, the providers of primary healthcare in many developed countries, require better support to address the needs of these families. This study engaged with CFHNs and healthcare interpreters to co-design a model of support for practice using workshops that included individual and collective brainstorming and visual representations. Transcripts of the discussion were analysed using thematic analysis. CFHNs and interpreters were able to articulate their perfect service model: a central multidisciplinary team of CFHNs, interpreters, and bilingual educators who could facilitate nurse–interpreter and nurse–interpreter–client relationships, allowing CFHNs and interpreters to do their jobs properly. This central structural component would support and be supported by rapport, trust, client choice and access, continuity of care, and cultural comfort. The study concluded that CALD mothers’ access and engagement require CFHNs to have support for their cultural comfort through the mechanism of bilingual educators and the expansion of healthcare interpreters’ role and scope in working with CFHNs in the delivery of services.

## 1. Introduction

Culturally and linguistically diverse (CALD) mothers with young children with limited English proficiency (LEP) experience greater inequities and face multiple impediments in accessing and engaging with primary health services compared to the host populations [1,2,3]. In the Australian context, ‘culturally and linguistically diverse’ is a widely used term that refers to those from non-English-speaking countries with specific cultural practices and values.

In Australia, as in other Western countries such as the United Kingdom and New Zealand, the main primary care provision for families with young children is through child and family health nursing services delivered by specialist child and family health nurses (CFHNs) (also known as health visitors, primary health nurses, and Plunket nurses). These services are designed and implemented using relationship-based practice to promote positive maternal and child health outcomes for mothers with children aged between birth and five years [4,5].

Previous research has identified that, despite the existence of child and family health nursing services, there are several factors that affect CALD mothers’ access to and engagement with these services [2,6,7,8]. These factors include language barriers, cultural incompatibility, social isolation, low health literacy, a lack of knowledge of healthcare services, and transport issues [3,9,10,11]. For example, an Australian qualitative study conducted with CFHNs identified barriers such as communication and cultural matters, mother–nurse trust relationships, and issues around working with healthcare interpreters [12]. Furthermore, these barriers continue impacting the well-being of CALD mothers and their young children, exposing them to a higher risk of negative health outcomes, including postnatal depression, pregnancy complications, and maternal and neonatal mortality [13,14].

There are also several facilitators of service access, including being connected to a community worker or bilingual staff and attending supported playgroups targeted to CALD women [15,16]. In addition, accessing appropriate gender-sensitive interpretation services (i.e., using female interpreters) has been shown to be a crucial factor in enhancing the engagement of CALD mothers with services. It has also been suggested that child and family health nursing services could better support and facilitate CALD families’ engagement with the services through an emphasis on continuity of care and building a mother–nurse trusting relationship [12].

Evidence suggests that using professional interpreters enhances trust within nurse–mother relationships and facilitates a better understanding of CALD mothers’ cultural practices [12,17]. However, Bonakdar Tehrani et al. [18] recognised that healthcare interpreters have ongoing challenges working with CFHNs in the delivery of child and family health nursing services to CALD families. These include factors such as a lack of sufficient time for pre- and post-briefing sessions with CFHNs and uncertainty about the interpreters’ role beyond language translation.

The persistence of CALD families’ difficulties accessing and engaging with CFHN services suggests that there is a need to develop a service model to support CFHNs and interpreters to better address the needs of these families. The aim of this study was to work with CFHNs and healthcare interpreters to co-design a model of child and family health nursing support for CALD families. This paper describes the challenges discussed and solutions developed through the co-design process.

## 2. Materials and Methods

### 2.1. Study Design and Context

This qualitative study used a co-design and participatory action research (PAR) approach to gain insights from CFHNs (service providers) and healthcare interpreters (service supporters) to develop a model of child and family health nursing support for CALD families with LEP (service recipients). The PAR approach is a collective approach based on self-reflective inquiry that enables researchers to work collaboratively with communities as active and powerful agents to understand and improve the world by changing it [19]. The PAR approach was used in this study to integrate and enable CFHNs and healthcare interpreters to engage in co-designing a model of child and family health nursing support that could be responsive to the needs of CALD mothers with LEP.

Co-design is more than a consultation process that has the potential to transform health and other public services. Co-design is a meaningful way to promote the involvement of end users in health research that reflects the voices and contributions of people with lived experiences [20]. Co-design is a collaborative interdisciplinary methodology that is being applied more frequently to improve healthcare systems and services [21]. Therefore, co-design was applied because it offered a solution-based approach and has been used effectively in healthcare improvement. The co-design process must consider the following dynamics: share power, prioritise relationships, use participatory means, and build a shared understanding through elevating the voices and contributions of participants with lived experiences [20,21]. In the context of this study, the end users are CFHNs (service providers) and healthcare interpreters (service supporters) who have lived experience in providing and supporting service delivery to CALD mothers.

### 2.2. Setting and Participants

The study took place in urban Sydney, Australia. This site is located in an area with a large CALD population. Health service policy within this area states that when a client requires or requests an interpreter, this function must be provided by a professional healthcare interpreter.

Initially, service managers were approached by the first author, who discussed the purpose of the study with them. Participants from two groups (CFHNs and healthcare interpreters) were then invited by service managers to participate. The service managers sought the verbal consent of CFHNs and healthcare interpreters to pass their contact details (email) to the first author. Subsequently, this convenience sample of participants was contacted by the first author via email to provide further information about the study, including the participant information sheet and consent form, and to arrange a date, time, and venue for the participants to attend the co-design workshops. The decision to conduct separate focus groups with CFHNs and healthcare interpreters was driven by pragmatic considerations, specifically to accommodate the diverse and demanding schedules of both participant groups, ensuring optimal engagement and meaningful contributions.

### 2.3. Ethical Consideration

The study was nested within a larger project that received ethics approval from the SWSLHD Ethics Committees (HREC/17/LPOOL/493). Voluntary written and verbal consent was obtained from all participants. Although invited by their managers to participate, participants were free to refuse to participate once contacted by the researchers. Due to the small sample, no demographic or professional data were collected to protect participant anonymity.

### 2.4. Data Collection

All nurses and interpreters who volunteered to participate attended the workshops. Overall, three CFHNs (all female) and five healthcare interpreters (four females and one male) participated in two separate workshops. The workshops were conducted in the participants’ workplaces and facilitated by two researchers (LK and MBT), both of whom were nurses. The workshops were conducted in July and August 2022. All written consents were received from the participants by the first author before the co-design workshops commenced. The duration of the workshops was between 60 and 90 min. All participants agreed to the workshops being audio recorded. Participants were viewed as experts in their domain, and as such, they had something to offer in designing a model and solutions. The four core principles of the co-design process—consent-driven, visual and creative, embodied, and connected to place—were implemented in convening the co-design process [20].

The co-design workshops aimed to encourage participants to discuss their thoughts and feelings about existing challenges in delivering child and family health nursing services that needed to be addressed. A summary of the findings from previous studies with CFHNs, healthcare interpreters, and CALD mothers [6,12,18], along with the existing literature [1], was used to prompt an initial discussion. Participants were then presented with broad questions such as ‘In a perfect world, what might be an ideal child and family health nursing model of support for CALD families with LEP to access and engage with the services?’ and ‘What would this model look like, and what things would it need to be perfect?’. Participants individually brainstormed, took notes, and then discussed collectively to synthesise their ideas. The co-facilitator (LK) captured the synthesised content on flipcharts as a visual representation during the workshop. At the conclusion of the workshop, the participants were asked whether the visual representation accurately reflected their views, and further modifications were made until all participants were satisfied that their views were reflected.

### 2.5. Data Analysis

Inductive reflexive thematic analysis, informed by Braun and Clarke [22], was applied to the visual representations and group discussion transcripts to integrate the CFHN and healthcare interpreter workshops into a single service design incorporating the themes and elements of both. Reflexive thematic analysis is a flexible method used to analyse qualitative data through an iterative process of identifying, analysing, interpreting, and reporting patterns or themes within a qualitative dataset [22]. The significance of this approach is that the subjectivity of the researcher is viewed and valued as fundamental to the analysis [23].

Initially, the first author familiarised and immersed herself with the data by reading and re-reading the visual representations and transcripts, listening to the audio recordings, and reading additional field notes several times. Then the first author inductively coded the visual representations and transcripts and generated initial themes. To ensure the reflexivity and rigour of the study, the data analysis was conducted independently by the research team (MBT, LK, and ST). Six phases of reflexive thematic analysis, as described by Braun and Clarke [22], were used, including (1) immersing and familiarising oneself with the data, (2) generating initial codes, (3) constructing (initial) themes, (4) reviewing potential themes, (5) defining and naming themes, and (6) producing and presenting the report. Consensus on the model of support for nurses working with CALD families was achieved through several online and face-to-face discussions and negotiations between the research team (MBT, LK, and ST). The socioecological framework was used to describe the elements of the co-designed model [24].

## 3. Findings

Through co-design workshops, a model of child and family health nursing support was designed for the delivery of child and family health nursing services to CALD mothers with LEP to support their access to and engagement with the services.

### 3.1. Co-Design Workshop with Child and Family Health Nurses

A visual representation of the ‘perfect service model’ developed by CFHNs is shown in Figure 1. Through the workshop with CFHNs, all participants agreed that CALD families should have a choice for the best match with a CFHN when receiving child and family health nursing services. The CFHNs believed that ensuring the best family–nurse cultural match would decrease the risk of disengagement of families from the services. The CFHNs highlighted that receiving e-learning cultural training could dispel cultural judgments. In addition, the CFHNs considered that cultural understanding and providing culturally specific resources are important to increase the comfort of delivering services to CALD mothers with LEP. The CFHNs also mentioned that access to culturally specific resources would enhance their knowledge, confidence, and surety in their mother–nurse relationship, which would improve the engagement of CALD mothers with the services.

The CFHNs discussed having onsite healthcare interpreters to bridge the long waiting time and improve responsiveness in the delivery of services to CALD mothers. They also indicated that ‘bilingual educators’ could assist in cultural understanding, which would create comfort and increase CFHNs’ confidence when working with this specific population. Furthermore, all CFHNs stated that, in the absence of providing onsite healthcare interpreters to work with CFHNs, a bilingual educator could facilitate the access of CALD mothers to child and family health nursing services. They also indicated that general practitioners (GPs) are an important source of information to provide instructions and encourage CALD families to access child and family health nursing clinics and services. All CFHNs suggested that having a pool of trained CFHNs, especially for high-risk clients (e.g., those who experience domestic violence), and providing translated resources are important to improve families’ engagement. In addition, CFHNs reported that access to appropriate local child and family health nursing clinics should be considered for mothers because, at times, mothers were reluctant to travel a long distance to a child and family health nursing clinic that only provided specific language interpreters. The CFHNs also indicated that the provision of telehealth appointments and booking an appropriate time based on mothers’ preferences should be considered.

Overall, all CFHNs agreed that a bilingual educator could increase continuity of care by building trusting mother–nurse relationships and decrease the risk of mothers’ disengagement from child and family health nursing services.

### 3.2. Co-Design Workshop with Healthcare Interpreters

A visual representation of the ‘perfect service model’ developed by the healthcare interpreters is shown in Figure 2. Although the healthcare interpreters generally stated that the system is good in terms of providing services beginning with pregnancy and continuing until the mother gives birth, there are too many barriers, such as a lack of bilingual healthcare providers. The healthcare interpreters stated that each family is unique, and a lack of time for briefing/debriefing sessions with CFHNs remains an issue. The healthcare interpreters reported that allocating enough time for briefing/debriefing sessions could increase their confidence, particularly for interactions with some specific cultures.

Furthermore, all healthcare interpreters agreed on the importance of having briefing/debriefing sessions for vulnerable families. They stated that briefings help interpreters understand how to approach clients (e.g., the way they say things, look, and use their body language). The healthcare interpreters could also reflect families’ concerns and hesitations identified during the session by having a debriefing with CFHNs. The interpreters believed that it would be helpful for healthcare providers to check on the client’s understanding and feelings during the session. In addition, they suggested that having a choice to access the same interpreters (e.g., starting in pregnancy) for CALD mothers could be a key element in building rapport and trust in interpreters. They also highlighted that a phone interpretation service instead of a face-to-face service is not appropriate or acceptable for most vulnerable families (e.g., refugees and families with LEP).

The healthcare interpreters also described the important role of education in relation to families’ cultural practices for healthcare professionals and in educating families as a result of literacy issues (e.g., poor attendance of mothers in group classes around gestational diabetes). All healthcare interpreters felt that they successfully performed in their interpreter role, even though they did not have access to the client’s medical history or knowledge of their background. All participants stated that their role sometimes goes beyond their scope of work, which is to facilitate family–nurse communication. They believed that they should be more involved in providing care because they felt that they could have a broader role (e.g., a cultural role) than being a literal translator.

Overall, all of the interpreters agreed that one size does not fit all. They suggested that working in a multidisciplinary team and having access to their client’s medical history could be beneficial because time spent retelling their story could be better spent building mother–nurse–interpreter rapport.

### 3.3. Service Design

The final developed model of support for child and family health nursing practice (see Figure 3) was designed by conducting a reflexive thematic analysis of the two models and transcripts from the two workshops. The central structural service component of the model is a multidisciplinary team that comprises relationships between the CFHNs, professional healthcare interpreters, and bilingual educators who both facilitate the CFHN–interpreter relationship and directly support the CFHN. Both groups felt that the current service model did not support them to do their jobs ‘properly’. The interpreters felt excluded from the multidisciplinary team and desired a greater understanding of the broader context of child and family health nursing service delivery. They highlighted ensuring their access to clients’ information and medical history would be beneficial and assist them in collaborating effectively with CFHNs. The interpreters recognised each family is different, and the allocation of time to learn about the family through pre- and post-briefing sessions with CFHNs would support them in approaching the clients with confidence and ‘doing the job properly’. Similarly, the CFHNs felt that they often did not have the level of training, knowledge, support, and service structures to provide the culturally safe environment for families that they wished. They felt that having the support of bilingual educators would enable the confidence and surety required for optimal service provision for CALD families.

The six themes, which are both the mechanisms and outputs of the model that facilitated ‘doing the job properly’, are described below.

#### 3.3.1. Choice

CFHNs and healthcare interpreters agreed on the importance of providing families with a choice to ensure the best family–nurse cultural match, onsite interpreters for ease of access in receiving child and family health nursing services, and more responsive service delivery. In addition, the healthcare interpreters emphasised mothers’ right to ask for the same interpreter to facilitate building rapport in mother–interpreter relationships.

#### 3.3.2. Rapport

Enabling the choice of a culturally matched CFHN and consistent use of the same interpreter facilitates rapport. It was noted by the CFHNs that cultural matching was not always possible, and the interpreters noted there was a need for more bilingual healthcare professionals. Despite this, the participants suggested that rapport could be enhanced by: ensuring families’ access to a pool of CFHNs specifically trained in the delivery of services to at-risk CALD mothers (e.g., domestic violence cases); providing on-site interpreters; interpreters having access to their client’s medical history; providing in-person services instead of phone interpreting services; and honouring mothers’ preferences for the same interpreter. These components were identified as key to building mother–nurse–interpreter rapport.

#### 3.3.3. Continuity

Both the CFHNs and interpreters highlighted the importance of continuity of care and that this would facilitate rapport and building relationships between the mother, nurse, and interpreter. The CFHNs, however, felt that the engagement of a bilingual educator could enhance the mother–nurse–interpreter relationship, recognising that the CFHN service had limited cultural diversity. The interpreters, however, emphasised that being involved in a multidisciplinary team and having access to the client’s medical history were the keys to facilitating continuity of care (noting that in the studied service, participating interpreters did not have access to clients’ records).

#### 3.3.4. Trust

The CFHNs agreed that bilingual educators could be the mechanism to bridge the communication and cultural gaps between nurses and families and enhance the nurses’ ability to build trusting relationships. Trust, both between the interpreter and the families and between the interpreter and the nurse, was viewed as important and could be facilitated by accessing the same interpreters and through interpreters being included as part of the child and family multidisciplinary service team.

#### 3.3.5. Access

The CFHNs emphasised the significance of mothers’ access to appropriate local child and family health nursing clinics, addressing the logistical challenges of long travel distances to clinics that exclusively provide specific language interpreters. They also highlighted providing telehealth appointments and considering mothers’ preferences for booking appointment times. The healthcare interpreters also underscored the importance of considering access to a face-to-face service, rather than just phone interpreting, as more appropriate and acceptable for most migrants and refugee families with the added vulnerability of LEP.

#### 3.3.6. Cultural Comfort

The CFHNs and interpreters emphasised the importance of having cultural understanding and confidence, providing culturally specific resources, collaborating with cultural leaders, accessing translated documents and resources, collaborating with bilingual educators, and access to cultural training. Participants suggested that these actions and resources were useful for dispelling judgement and avoiding cultural assumptions. Both participant groups noted the importance of acquiring knowledge about each family’s specific and individual expressions of culture in order to enhance the service providers’ understanding of and approach to the mothers’ cultural practices. All of these would support creating and enhancing the service providers’ comfort in the delivery of the services to CALD mothers.

## 4. Discussion

The co-design research approach provided opportunities for the study participants (CFHNs and healthcare interpreters) to share their knowledge and experiences of providing child and family nursing services to CALD families with limited English proficiency, requiring the engagement of interpreters. The highlight of the proposed model is how the role of a bilingual educator—a core element of both the CFHN and healthcare interpreter models—can facilitate nurse–mother interactions directly, as well as nurse–mother–interpreter interactions. There is a need to build the capacity of healthcare staff (CFHNs and healthcare interpreters) and systems to work with and care for CALD families with LEP in a collaborative and multidisciplinary team. The focus of a multidisciplinary team approach to care is to ensure that CFHNs and healthcare interpreters can engage and plan care together. To achieve this, healthcare interpreters require access to clients’ data, information, and medical history so they can work efficiently with nurses to ‘do[ing] the job properly’. This study identified that ‘doing the job properly’ also meant providing services in the context of relationships that facilitated rapport, choice, cultural comfort, continuity of care, access, and trust.

Being involved in a multidisciplinary team can facilitate and enhance the continuity of care by building rapport and a trusting mother–nurse–interpreter relationship. This is supported by previous research that showed that building a consistent relationship between the nurse, interpreter, and family is a key component for building trust and continued engagement in the delivery of the services [12,16]. The role of the bilingual educator could facilitate and positively reinforce the practice concepts identified in both workshops—rapport, choice, continuity of care, access, and trust—which have been well recognised in the literature as needed for working effectively in a relationship-based practice [5,12,14]. These benefits were also consistent with what CALD mothers have revealed as important and supportive factors in accessing and engaging with the services [6].

This research identified that CFHNs would like more support to understand mothers’ cultural practices and provide cultural resources to create comfort and confidence in service provision for both themselves and the mothers engaging in the services [1,12,18,25,26]. The concept of cultural ‘comfort’ as expressed by the nurses and the approach described by interpreters extends beyond practitioner knowledge of, skills working with, or attitudes toward the cultural practices of particular diverse groups to ways of engaging in individual relationships with their clients [27]. Cultural comfort is a concept used to describe the ways clients and therapists interact, co-create, and experience their relationships in the context of cultural dynamics [28]. It has been recently identified in the literature relating to psychotherapy practice but has not yet been applied to child and family health nursing. It is characterised by the service provider as “feeling at ease, open, calm, or relaxed with diverse others” ([28], p. 92, [29,30]), and has been shown to impact client retention and cultural safety and to promote a more open clinical environment and reduce client distress [30,31]. This finding is consistent with the previous qualitative study, which identified that, in the absence of support for nurses to be culturally comfortable and provide a culturally safe practice, CALD mothers using child and family health nursing services had to manage their cultural matters and at times hide their cultural practices [6], that is, practice cultural concealment [32].

In this study, the bilingual educator was the mechanism identified by the participants as best supporting the identified elements of the relational practice (rapport, choice, continuity of care, access, and trust) and the cultural comfort of the whole team, facilitating and supporting effective working within the team. This resonates with Davis et al.’s [28] review of cultural comfort, which noted the need for cultural consultants as one of the supportive mechanisms. In this way, a bilingual educator could assist CFHNs in understanding cultural matters and increase CALD mothers’ willingness to share their cultural practices in parenting, which could help in accessing and engaging with services. Consistent with the findings of this study, the value of having community health workers such as bicultural or bilingual workers to improve the health of the population, and especially of vulnerable populations, has been identified in previous studies, both nationally and internationally [15,25,33,34,35,36,37,38]. Furthermore, research has demonstrated the crucial role of community health workers in improving maternal and child health in low- and middle-income countries (e.g., exclusive breastfeeding, reduced neonatal mortality, improved mother–infant bonding, and improved children’s cognition) [38,39]. While the effective role of community health workers is well recognised in low- and middle-income countries, the lack of sustainable investment in recognition of their role in high-income countries, especially for vulnerable and marginalised populations, is evident [40].

### Strengths and Limitations

This study only involved a small number of providers within one health service, albeit one with a large CALD population. Although CALD mothers with LEP in other high-income countries may have access to different child and family and maternity healthcare systems, using this co-designed model of child and family health nursing support for practice with CALD families could increase cultural competency and comfort and be transferable to other countries with similar services.

The current study did not trial and evaluate the effect of the developed model of child and family health nursing support on the practice of CFHNs and healthcare interpreters, nor on the knowledge of CALD mothers. Neither does the model provide operational service guidance on how such a multidisciplinary team can be organised or such practical implementation considerations as ratios of nurses to bilingual educators or interpreters. However, the findings of Australian studies [16,27] support the effectiveness of having bicultural workers to support women in maternity care, bridge communication gaps, and build trusting relationships with healthcare professionals. Nevertheless, neither of the previous studies evaluated the outcomes to determine whether their models could result in better health outcomes for this particular population [15,25]. Future research should explore mechanisms for operationalising the model and measure the impact of the developed model on CFHNs’ cultural comfort and the outcome for families.

## 5. Conclusions

The service model developed in this study was co-designed with CFHNs and healthcare interpreters, which allowed them to collaborate in dynamic and interactive workshops to develop a model of support for child and family health nursing that has the potential to be acceptable, practical, and responsive to the needs of CALD mothers with young children who have the added vulnerability of LEP. The study identified the need for the provision of greater support for the cultural comfort of child and family health nurses through the mechanism of bilingual educators and expanding healthcare interpreters’ role and scope in working with CFHNs in the delivery of the services. Having systematic sustainable cultural support within early childhood services could facilitate improved and effective family–nurse–interpreter relationships and potentially address and improve service engagement and outcomes for CALD families.

## Figures and Tables

**Figure 1 ijerph-21-01274-f001:**
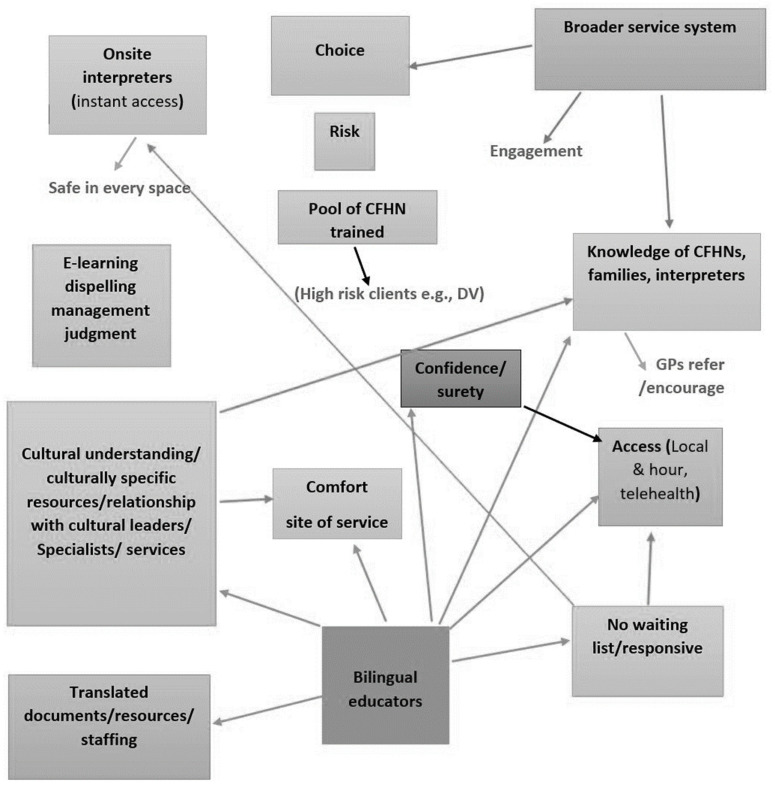
Visual representation of the ‘perfect service model’ developed by the CFHNs (shading for ease of reading only).

**Figure 2 ijerph-21-01274-f002:**
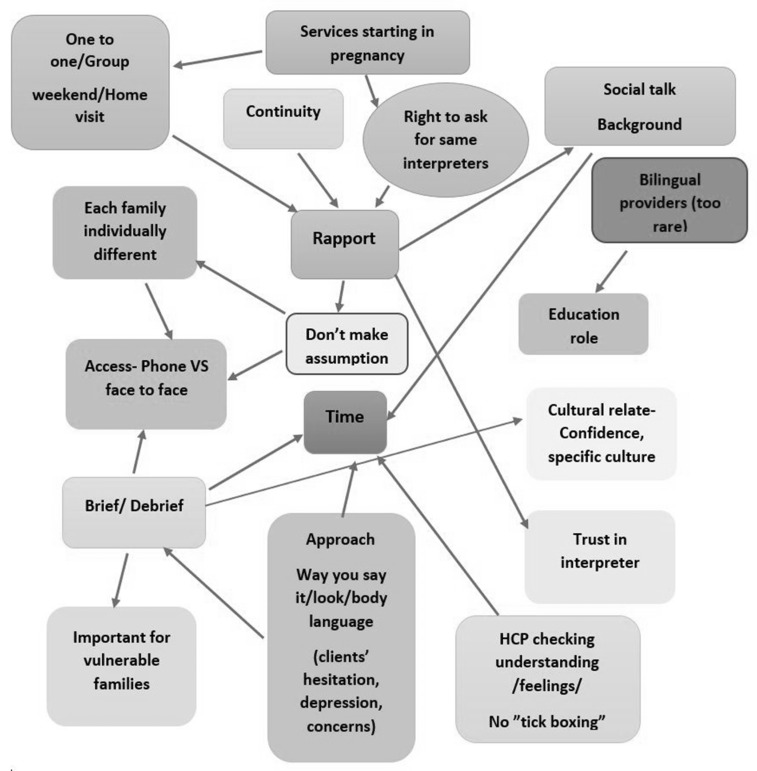
Visual representation of the ‘perfect service model’ developed by healthcare interpreters (shading for ease of reading only).

**Figure 3 ijerph-21-01274-f003:**
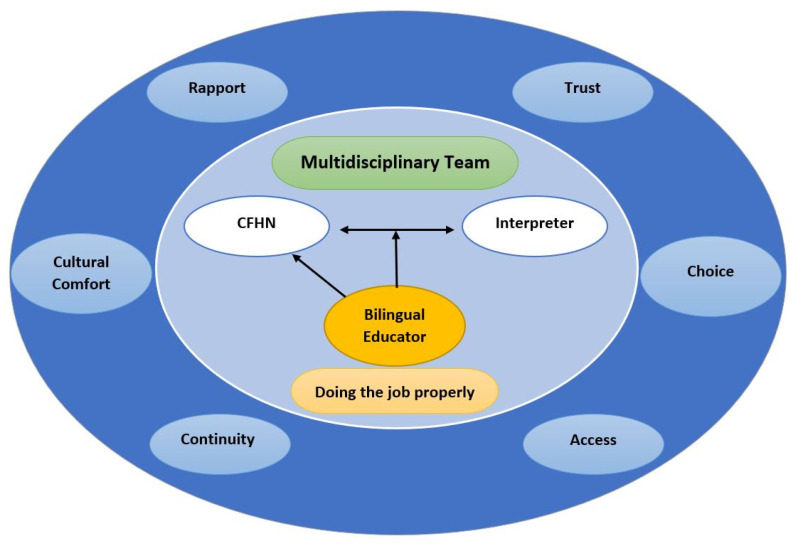
Model of support for child and family health nursing services for CALD families.

## Data Availability

The data presented in this study are available on request from the corresponding author due to ethical requirements to maintain participant privacy.

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
