# Peer review of "Co-Design Model of Support for Child and Family Health Nurse Practice with Culturally and Linguistically Diverse Families"

_ijerph, 2024, doi:10.3390/ijerph21101274_

Round 1
Reviewer 1 Report
Comments and Suggestions for Authors
The article "Co-Design Model of Support for Child and Family Health Nurse Practice with Culturally and Linguistically Diverse Families" was considered for review. The authors aimed to "work with CFHNs and healthcare interpreters to co-design a child and family health nursing support model for CALD families." The article has scientific merit and is well-written. The solidity of the method is highlighted. Despite this, some recommendations are indicated:
• Although the authors indicated the focus group conductors (by code), mentioning the researcher's credentials (nurse?) is essential.
• Another critical point is to determine whether a relationship was established before the start of the study and whether the participants were aware of the researchers/workshop facilitators (e.g., reasons for developing the research).
• In the method, although the researchers pointed out how they approached the study participants, it is essential to mention the type of sampling (convenience, consecutive, snowball?).
• It was also unclear whether there were refusals since only 8 representatives participated.
• It is also necessary to address where the co-design workshops were held (in the clinic and workplace?).
• Important characteristics of the sample, such as demographic and professional data, were also not described, and the collection date was not included.
• Another point not mentioned is whether the transcripts were returned to the participants for comments and correction.
• In the Discussion, the researchers presented excellent reflections on "Cultural Comfort," one of the "six themes that are both mechanisms and outcomes of the model, which facilitated 'doing the job right'" (lines 257-258, page 8 of 13), described in the Results section. However, the other items (Choice, Rapport, Continuity, Trust, and Access) were addressed in an incipient manner. It is suggested that the discussion of these themes be expanded, including the definitions adopted by the manuscript's authors.
Reviewer 2 Report
Comments and Suggestions for Authors
The topic of this study is very good and is suitable for attracting readers' attention. However, the description of the study needs to be supplemented. 1. Research ethics - IRB approval alone does not prove ethics. Please describe this in detail. 2. It seems that the integrated model only has professional interpreters. Please present the integrated model that the researcher thinks of. 3. It is said that inductive reflexive thematic analysis was used, but there is no information about this in the research results. In other words, since there is no information presented at the Co-Design Workshop, it is not known how this result was derived and whether it was agreed upon at the workshop. Also, how are the personnel organized in the integrated model, and how many interpreters are needed? How were they organized as a team and what roles were assigned? Is there an existing team? This medical model will vary by country and region, and since I do not know what model is used in this region, I do not understand the model. The content should be revised to understand the readers. 4. The research results seem to be rapport, trust, access, etc. that this model should show rather than an integrated model. How is this element different from the previously presented elements? This part should be presented in the discussion, but is it the content derived from the research results? It needs to be revised.
Round 2
Reviewer 2 Report
Comments and Suggestions for Authors
The authors have thoroughly addressed all of the reviewer's comments and have made appropriate revisions.